# The Consistency Factor and the Viscosity Exponent of Soybean-Protein-Isolate/Wheat-Gluten/Corn-Starch Blends by Using a Capillary Rheometry

**DOI:** 10.3390/molecules27196693

**Published:** 2022-10-08

**Authors:** Wei Zhang, Donglin Zhao, Ziyan Dong, Jian Li, Bo Zhang, Wenhua Yu

**Affiliations:** 1Institute of Food Science and Technology, Chinese Academy of Agricultural Sciences/Key Laboratory of Agro-Products Processing, Ministry of Agriculture and Rural Affairs, Beijing 100193, China; 2Beijing Advanced Innovation Center for Food Nutrition and Human Health, Beijing Technology and Business University, Beijing 100048, China; 3Research and Development Department, Shandong Wonderful Biotech Co., Ltd., Dongying 257500, China

**Keywords:** soybean-protein-isolate, corn-starch, wheat-gluten, consistency factor, viscosity exponent, capillary rheometry

## Abstract

Blends with different proportions of protein or starch show different rheological behaviors, which may be related to the fibrous structure formation of extruded textured plant proteins. The consistency factor *K* and the viscosity exponent *n* of soybean–protein–isolate (SPI)/wheat–gluten (WG)/corn–starch (CS) blends were investigated through capillary rheometry. All blends exhibited shear-thinning behavior at 80 °C and 50% moisture. The CS content in SPI/CS blends or WG content in SPI/WG blends showed a positive relation to the viscosity exponent *n* and a negative relation to the consistency factor *K*. However, there was no correlation between the CS content in WG/CS blends and *n* or *K*. The coefficient of determination of the linear relationship between *K* and mass fraction in SPI/CS, SPI/WG/CS, SPI/WG and WG/CS decreased from 0.872 to 0.073. SPI was more likely to form a non-interactive structure, while wheat-gluten was more likely to form a highly interactive structure. It turned out that the materials with globular morphology, such as soybean-protein-isolate and corn-starch, are likely to form a non-interactive structure.

## 1. Introduction

Plant protein, induced by extruded mechanical energy and thermal energy, can form meat-like products with a macroscopically fiber-like structure, which is considered to be an important quality characteristic of plant-based meat analogs [1]. The laminar flow state of the material at the die forms a velocity gradient similar to the Hagen–Poiseuille flow, which is fast in the middle and slow on both sides, resulting in the formation of a fiber-like structure with a certain orientation macroscopically [2]. Therefore, the fibrous structure is affected by the material’s flow behavior. The flow behavior of materials can be modeled by the power law function of shear rate and shear stress, $\dot{\gamma }$ = *ϕ* · *τ^m^*, where the *m* value is the flow exponent, $\dot{\gamma }$ is the shear rate, τ is the shear stress and ϕ is the fluidity. It is found that the *m* value affects the velocity gradient at the die head, and with an increase in the *m* value, that is, the pseudo-plasticity, the velocity distribution presents a compressed flow profile, and with a decrease in the *m* value, the velocity distribution shows an extensional flow profile [3]. Additionally, according to the power law model of shear viscosity and shear rate, $\eta =K{\dot{\gamma }}^{n-1}$, where η is shear viscosity, K is the consistency factor, $\dot{\gamma }$ is the shear rate, *n* is the viscosity exponent and *n* = 1/*m*. Stretching natural polymer proteins need to overcome entropy elasticity; that is, the product (*S* × τ) of the stretching velocity gradient (*S*) and the relaxation time (τ) of the molecular chain contraction should be large enough so that the protein could be stretched. The relaxation time τ is proportional to the square of the system viscosity [4]. The *K* value is the consistency factor and is correlated with the relaxation time. *K* expresses the viscosity at the shear rate $\dot{\gamma }$ ~ 0. The higher the *K* value, the higher the viscosity.

The viscosity exponent *n* and consistency factor *K* of protein or starch are not only affected by moisture and temperature [5,6] but also the type of materials. When pea protein isolate accounted for 90% and the remaining 10% was composed of amylose and amylopectin, with the increase in amylopectin from 0 to 10% and the decrease in amylose from 10% to 0, the proportion of amylopectin is negatively correlated with the *K* value (R^2^ = 0.9931), and is quadratically correlated with the n value (R^2^ = 0.9934) [7].

The mixed materials in the phase separation or compatible state have different flow behaviors. The flow behavior of whey protein isolate and cross-linked waxy corn–starch blends exhibit that, with the increase in the mass fraction of cross-linked waxy corn–starch from 0 to 1, the G’ of blends indicate the range of phase compatible states. When the mass fraction of cross-linked waxy corn–starch is between 0.4 and 0.8, the G’ is relatively low, indicating the system may be composed of two continuous phases in a compatible state; when the mass fraction of cross-linked waxy corn-starch is less than 0.4 or higher than 0.8, the G’ is relatively low, indicating the system is filled by dispersed cross-linked waxy corn-starch particles in the continuous whey protein network, or the aggregated protein particles fill the space between the gelatinized starch particle matrix, and the system is in a state of phase separation [8].

In the plant protein extrusion process, protein and starch are the main components, and each of those may form the continuous phase [8]. With the change in the proportion of components, there will be different phase separations, compatibility behaviors and *K* and *n* values, which may be related to the fibrous structure formation of extruded, textured plant proteins. In this study, soybean–protein–isolate, wheat–gluten and corn–starch, which are commonly used in plant protein extrusion, were used as experimental materials to study the consistency factor *K* and the viscosity exponent *n* of the protein–starch blends with different mass fraction components. The degree to which the relationship between flow behavior and mass fractions deviated from the non-interactive model was evaluated. This helps in understanding and controlling the rheological behavior of protein–starch blends and may further control the formation of the fibrous structure.

## 2. Results

### 2.1. DSC of Soybean-Protein-Isolate, Corn-Starch, and Wheat-Gluten

The DSC results of soybean–protein–isolate (SPI), corn–starch (CS) and wheat-gluten (WG) are shown in Figure 1 and Table 1. The peak temperature of phase transition (T_p_) for SPI and CS (both with 50% moisture) was 106 °C and 68 °C, respectively, while there was no distinct thermal transition peak for WG. This indicates that 80 °C is not enough to totally denature the SPI; the CS probably was in a partially gelatinized state, and the WG has been completely denatured.

### 2.2. Rheological Characterization of Soybean-Protein-Isolate/Corn-Starch Blends

Viscosity as a function of shear rate for different soybean–protein–isolate (SPI)/ corn–starch (CS) blends with 50% moisture content at 80 °C barrel temperature is shown in Figure 2A. Measured viscosity values of all blends were observed to decrease with the shear rate increasing from 10 to 1000 s^−1^ and with CS content increasing from 0 to 100% in the blends.

Power law parameters as a function of CS content in the SPI/CS blends are shown in Figure 2B,C. The viscosity exponent *n* values increased from 0.297 to 0.464, indicating a marked shear-thinning behavior in the blends, and appeared positively correlated with CS content (R^2^ = 0.843). The consistency factor *K* values decreased from 119,492 to 3004 Pa·s. The *K* value of blends decreased with the increase in CS content from 0 to 100% and was negatively correlated with the CS content (R^2^ = 0.872).

### 2.3. Rheological Characterization of Soybean-Protein-Isolate/Wheat-Gluten Blends

Viscosity as functions of shear rate for different soybean–protein–isolate (SPI)/wheat-gluten (WG) blends with 50% moisture at 80 °C is shown in Figure 3A. Similar trends as those seen in SPI/CS blends were observed. The viscosity of all blends decreased as the shear rate increased from 10 to 1000 s^−1^ and as WG content increased from 0 to 100%.

The relation between power law parameters of different SPI/WG blends and WG content is shown in Figure 3B,C. The *n* value increased from 0.297 to 0.419, not only showing a shear-thinning behavior but also suggesting the formation of some network [9], as WG content increased from 0 to 100% in blends and was positively correlated with the WG content (R^2^ = 0.837). The consistency factor *K* decreased from 119,492 to 3,939 Pa·s with WG content increasing from 0 to 100%, and in a negative relationship with WG content (R^2^ = 0.575).

### 2.4. Rheological Characterization of WG/CS Blends

Viscosity as a function of shear rate for different WG/CS blends is shown in Figure 4A. The viscosity decreased when the shear rate increased. The viscosity of WG/CS blends was higher than WG or CS viscosity alone, totally different from the blends of SPI/CS and SPI/WG, where the blend viscosity was between SPI and CS or SPI and WG, respectively.

The power law parameters of different WG/CS blends are shown in Figure 4B,C. There was no correlation between the power law parameters and the ratio of the components in the WG/CS blend, unlike that of SPI/CS or SPI/WG blends, indicating some interaction occurred between WG and CS.

### 2.5. Rheological Characterization of the Blends of SPI/WG/CS

The viscosity of different SPI/WG/CS blends obtained at 50% moisture and 80 °C is shown in Figure 5. Figure 5A–C demonstrate the results obtained when the ratio of WG/CS was 1/3, 1, and 3, respectively. The viscosity of blends gradually increased with SPI content increasing from 0, 60, 80 to 100% when the ratio of WG/CS was 1 or with SPI content increasing from 0 to 60% when WG/CS was 3. However, there was hardly any difference in blends’ viscosity with SPI content ranging from 0 to 60% when WG/CS was 1/3, which may indicate a special structure in the WG/CS system.

The power law parameters of the different SPI/WG/CS blends are shown in Figure 5D,E. When the ratio of WG/CS was 1, the viscosity exponent *n* decreased from 0.404 to 0.246, and the consistency factor *K* increased from 6,087 to 119,492 Pa·s with increasing SPI content. The same trend was observed at 3/1 and 1/3 ratios for the consistency factor *K*, which increased from 10,837 to 119,492 Pa·s and from 4,101 to 119,492 Pa·s, respectively.

### 2.6. Fitted Equations between Mass Fraction and n_no-inter_ or K_no-inter_ of a Non-Interactive Model

The *η_no-inter_*, *n_no-inter_* or *K_no-inter_* of each blend were calculated according to the non-interactive model. The fitted equations between the mass fraction of components and *n_no-inter_* or *K_no-inter_* are shown in Table 2 and Table 3. The linear, exponential, power function, and logarithmic equations were used to fit the relations between mass fraction and *n_no-inter_*, in which the first two equations made sense. The decisive factors R^2^ of the linear fitted equation between the mass fractions of components and the *n_no-inter_* were 0.625, 0.631, 0.997, 0.881, 0.862 and 0.866, respectively, and the R^2^ values of the exponential fitted equations were 0.652, 0.652, 0.999, 0.885, 0.867 and 0.870, respectively, indicating that the relationship between the mass fraction of components and *n_no-inter_* was unstable under linear or exponential equations. This suggested that the viscosity exponent *n* of the blends was complicated. There was no simple model which could demonstrate the relationship between the mass fraction and the viscosity exponent *n*, even with the *n_no-inter_* of none interaction model.

As for the consistency factor *K_no-inter_*, the decisive factors R^2^ of the linear fitted equation were 0.9999, 1.0000, 0.9993, 1.0000, 1.0000, and 1.0000, respectively, while the R^2^ of the exponential fitted equation was 0.813, 0.828, 1.000, 0.931, 0.923, and 0.926 for each blend, indicating that linear equation was suitable for describing the relationship between mass fraction and *K_no-inter_*, for the R^2^ was stable and was almost 1.0000, which suggested the R^2^ of linear equations between mass fraction and measured *K* values may serve as the index presenting the interaction degree of components in the blend. In the linear equation between mass fraction and *K*, a further R^2^ from 1 indicates a higher degree of interaction.

## 3. Discussion

Protein and starch are the main components in plant protein extrusion, and each of them may form a continuous phase [8]. Blends with different proportions of protein and starch exhibit different phase separation behavior and rheological behavior with different *K* and *n* values, which might be related to the fibrous structure formation of extruded textured plant proteins.

To study a protein/starch blend’s rheological behavior, the consistency factor *K* and the viscosity exponent *n* of soybean-protein-isolate/wheat-gluten/corn-starch blends were investigated.

Reference [10] illustrates that *n* increases and *K* decreases as corn–starch (CS) content increases in soybean-protein-isolate (SPI)/CS blends, presenting similar trends to our results. Reference [11] found increasing *n* and decreasing *K* in SPI/wheat–gluten (WG) blends when WG increases, also presenting similar trends to our results. The authors of [12] studied the flow behavior of wheat–starch/WG blends under certain conditions (35% moisture content, 140 °C) and found that the viscosity exponent *n* increased and consistency factor *K* decreased with the increase in wheat–starch content, which is different from our results, and we believe this is because the wheat–starch had totally gelatinized at 140 °C and replaced gluten to become the dominant continuous phase in the blends [12,13].

Two different polymers will interact with each other when mixed, either by combining them together or by separating them [14,15]. To test the interaction degree, an ideal model with no interaction was constructed and used as a standard to compare with the measured values. The larger the difference between the non-interactive model values and the measured values, the higher degree of binary interaction in the blend. The relationship between the mass fraction of components and consistency factor *K_no-inter_* or viscosity exponent *n_no-inter_* of a non-interactive model was evaluated with linear, exponential, power function and logarithmic equations. Only the linear relationship between mass fraction and *K_no-inter_* was stable, and all the R^2^s of the equations were almost 1.000. Thus, the R^2^ of the linear equations between mass fraction and measured *K* values may act as the index presenting the interaction degree of components in the blend, where a further R^2^ from 1 indicates a higher degree of interaction. After the linear relationship between measured consistency factor *K* and the mass fraction of components was calculated and the deviation from R^2^ = 1 was evaluated, we found that the R^2^ of the relationship between measured *K* and mass fraction in SPI/CS, SPI/WG/CS (WG/CS = 3/1), SPI/WG/CS (WG/CS = 1/3), SPI/WG/CS (WG/CS = 1/1), SPI/WG, and WG/CS were 0.872, 0.785, 0.697, 0.597, 0.574, and 0.073, respectively, and the deviation from R^2^ = 1 was 0.128, 0.215, 0.303, 0.403, 0.426, and 0.927, respectively. Those may indicate that it is also the sequence of the increasing degree of the binary interaction.

Different from a low concentrated system, the degree of binary interaction in a highly concentrated system is related to the molecular morphology. SPI is mainly composed of globulins corresponding to storage proteins of soybeans, including 7S and 11S, whose thermal transforming temperature ranges between 89.97 °C(7S) and 108.79 ℃(11S) with 50% moisture [16,17]. The T_p_ of the SPI sample in the experiment with 50% moisture was 106 °C, which indicates that SPI retains a partially globular morphology under 50% moisture at 80 °C. The microstructure of gluten networks has revealed that the interacting and interpenetrating proteins could form a continuous network [18,19]. The glutenin fraction comprises aggregated proteins linked by inter-chain disulfide bonds [20]. Imaging techniques show strong evidence that gliadins interact at a molecular level with each other via physical forces, including hydrogen and ionic interaction [21]. The WG/CS blends with 50% moisture, similar to wheat flour dough, even to sourdough with low pH or waxy flour dough, where the starch is observed to be embedded in the continuous protein matrix and form a stable structure [22,23]. As for WG/SPI blends, the soy globular protein bodies are immersed in the gluten fibrils, interfering in the continuous gluten network [11]. It is possible that both binaries with globular morphology are not interactive. However, there is an equal possibility (which requires further investigation) that the introduced component with network morphology is likely to form a high degree of interactive structure.

## 4. Materials and Methods

### 4.1. Materials

Soybean-protein-isolate (SPI) was supplied by DuPont (Zhengzhou, China), with protein and water content of 89.43% (dry basis) and 5.79%, respectively. Corn-starch (CS) was supplied by Ingredien (Shanghai, China), with corn and water content of 99.38% (dry basis) and 11.85%, respectively. Wheat–gluten (WG) was supplied by Lianhua Group Ltd. (Zhoukou, China), with protein and water content of 77.30% (dry basis) and 7.49%, respectively.

### 4.2. Differential Scanning Calorimetry

The DSC curves were obtained using thermal analysis systems (Q-200, TA Instruments, New Castle, DE, USA). Samples were conditioned in hermetic aluminum TA pans with 50% moisture, weighed (8–10 mg) using a precision balance (±0.01 mg, Analytical Plus, Mettler Toledo), and heated at a rate of 10 °C min^−1^ between 20 and 130 °C under an inert atmosphere (50 mL min^−1^ of dry N_2_). The reference was a void aluminum TA pan. The onset temperature (T_o_), peak temperature (T_p_), and enthalpy (ΔH) were computed from the curves by the Universal Analysis Program, Version 1.9 D (TA Instruments).

### 4.3. Samples Preparation

The blends of SPI, CS and WG were made at the ratios listed in Table 4. According to the barrel, the moisture content was in the range of 40–60% during high moisture extrusion of plant-based protein, so the moisture content of 50% was chosen in all tests. Water was added to each protein–starch blend (as part of the total sample mass at a dry base) and mixed in a food cooker HR7633/10 (Philips, Holland). The water content of the protein and starch was determined and included in the total amount of water added to the dry blend. The moisture content of the blends was checked using the AACC method (1995). The conditioned samples were packed in plastic bags and kept at 4 °C overnight for equilibration.

### 4.4. Rheology Measurements

A capillary rheometry (Göttfert, RHEOGRAPH 25; Buchen, Germany) of a dual barrel system was used to measure the shear viscosity of the samples. Round hole capillaries with an inside diameter (1 mm) and length (10 mm) were used. The length/diameter (L/D) value of the selected capillary was 10. The capillary was fixed at the bottom of the barrel. Generally, during high moisture extrusion of plant-based proteins, the barrel temperature is higher than 100 °C, which will cause rapid moisture dissipation and bake the samples during rheology measurements. Thus, the temperature of 80 °C was selected, which was close to 100 °C and caused slow dissipation. The temperature was set at 80 °C along the barrel during the experiment. The barrel was filled with a similar quantity of each sample, and it took about 6 min to complete a filling. The piston was allowed to contact the sample in the barrel; the piston moved down and stopped at the first appearance of dough at the bottom of the capillary. Then, the sample was allowed to equilibrate for 5 min before starting the test. The experiments were performed at increasing apparent shear rates (10–1000 s^−1^) corresponding to increasing piston speeds. The pressure needed to extrude the sample through the capillary was recorded. The tolerance of the measuring pressure was set as 2%, and the pressure was recorded upon reaching a steady state within this tolerance.

The power law model, Equation (1), was used to fit the viscosity results of blends [24].
(1)$$\eta =\frac{\tau }{\dot{\gamma }}=K{\dot{\gamma }}^{n-1}$$
where η is the shear viscosity, τ is the shear stress, $\dot{\gamma }$ is the shear rate, *K* is the consistency factor, and *n* is the viscosity exponent.

There are Newton regions and non-Newton regions for all samples, with shear rates increasing from 0 to 1000 s^−1^. Using the more accurate models to fit the two flow behaviors will make the study complex; the universal power law equation was used to describe the rheology characterization. The power law equation and parameters (*n*, *K*) of samples were obtained via Origin 2022, and all the coefficients of determination (R^2^) of fitting models are no less than 0.9860. In detail, fit the shear rate and its corresponding viscosity data with the power law function and obtain *K* and *n* values using the nonlinear curve fit function of Origin 2022. With the power law model of each component, the viscosity data of each component at a certain shear rate can be calculated. To assess the repeatability of the capillary rheometry measurement, the measurements with 100% SPI, 100% WG, or 100% CS were repeated in separate runs. The power law equation and parameters (*n*, *K*) of three samples were repeatable, with variances less than 1%. Thus, for all other blends, one capillary rheometry measurement was carried out.

### 4.5. A Non-Interactive Model

The blend viscosity was supposed to be related to the proportion and the viscosity of each component when there was no interaction [25]. A non-interactive model, where the rheological properties were linearly correlated with the mass fraction of each component, was presumed. The value of blend viscosity from this non-interactive model was calculated in Equation (2)
(2)$$\eta_{no-inter}=\frac{W_{1}\eta_{1}+W_{2}\eta_{2}\cdots +W_{n}\eta_{n}}{W_{1}+W_{2}\cdots +W_{n}}$$
where *η_no-inter_* is the calculated blend viscosity, *W_n_* is the mass fraction of each component, and *η_n_* is the viscosity of each component at a certain shear rate, which is calculated with the power law model of each component.

Then, *K_no-inter_* and *n_no-inter_* were fitted from the power law model with *η_no-inter_* and the corresponding shear rate. These *K_no-inter_* and *n_no-inter_* of the non-interactive model were merely used as a standard to compare with the measured *K* and *n* values of all samples. The larger the difference between the measured values and the values from the non-interactive model, the higher the degree of interaction within the components of the blends.

### 4.6. Statistical Analysis

All the statistics in this study were calculated by PASW Statistics 18.0 software, and the equation fitting was achieved by origin 2022 (Origin Lab, Northampton, USA). The content scatter diagrams and the relationship between *K* or *n* and mass fraction of SPI/WG/CS were drawn and fitted by EXCEL. The relationship between the mass fraction of components and *K* or *n* included linear equations, exponential equations, power function equations, and logarithmic equations.

## 5. Conclusions

The rheological properties of different blends with different soybean–protein–isolate (SPI)/wheat–gluten (WG)/corn–starch (CS) ratios were investigated through capillary rheometry. The blends exhibited shear thinning behavior at 50% moisture and 80 °C, with the viscosity exponent *n* ranging from 0.245 to 0.466. CS content in SPI/CS blends and WG content in SPI/WG blends showed a positive correlation with the viscosity exponent *n* and a negative correlation with the consistency factor *K*. However, there was no correlation between the WG content and *n* or *K* in WG/CS blends. The coefficient of determination of the linear relationship between *K* and mass fraction in SPI/CS, SPI/WG/CS, SPI/WG and WG/CS decreased from 0.872 to 0.073. SPI is more likely to form a non-interactive structure, while wheat–gluten is more likely to form a highly interactive structure. It turned out that both binary and globular morphology, such as soybean–protein–isolate and corn–starch, are likely to form a non-interactive structure.

## Figures and Tables

**Figure 1 molecules-27-06693-f001:**
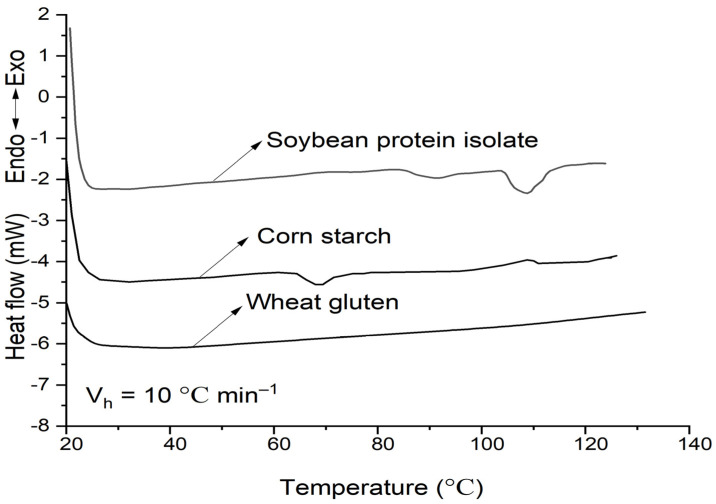
DSC curves of soybean–protein–isolate, corn–starch, and wheat–gluten. The moisture was 50%, and the heating rate was 10 °C min^−1^.

**Figure 2 molecules-27-06693-f002:**
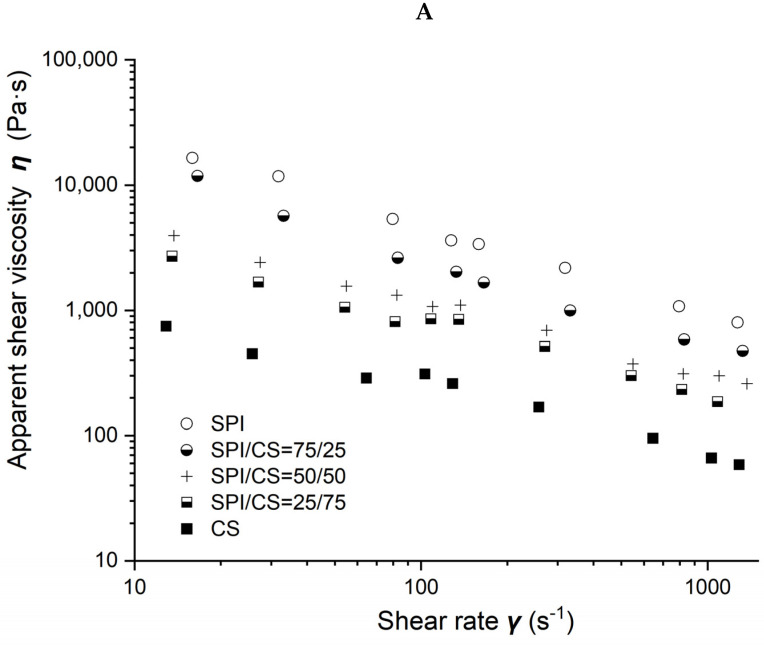
Rheological properties of soybean–protein–isolate (SPI)/corn-starch (CS) blends. Temperature was 80 °C and water content at 50% *w*/*w*. (**A**) Apparent shear viscosity; (**B**) consistency factor versus CS content in SPI/CS blends; (**C**) viscosity exponent versus CS content in SPI/CS blends.

**Figure 3 molecules-27-06693-f003:**
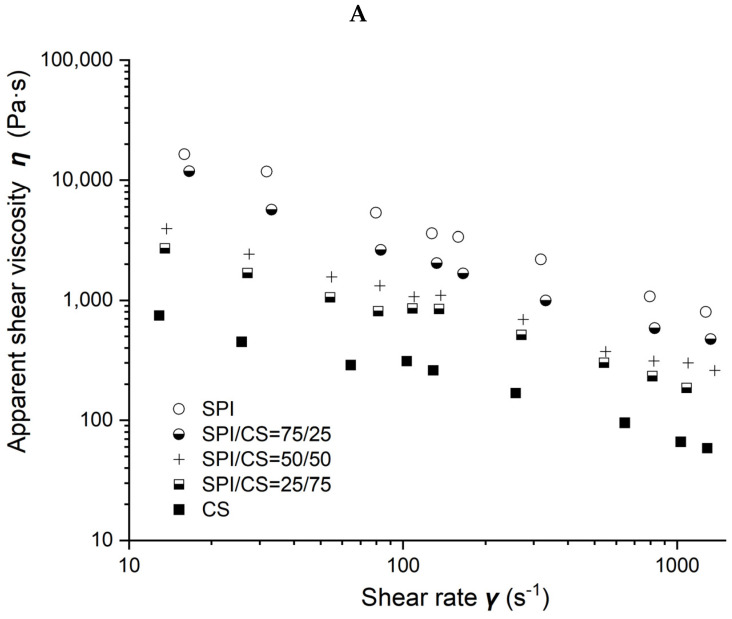
Rheological properties of soybean–protein–isolate (SPI)/wheat–gluten (WG) blends. Temperature was 80 °C and water content at 50% *w*/*w*. (**A**) Apparent shear viscosity; (**B**) consistency factor versus WG content in SPI/WG blends; (**C**) viscosity exponent versus WG content in SPI/WG blends.

**Figure 4 molecules-27-06693-f004:**
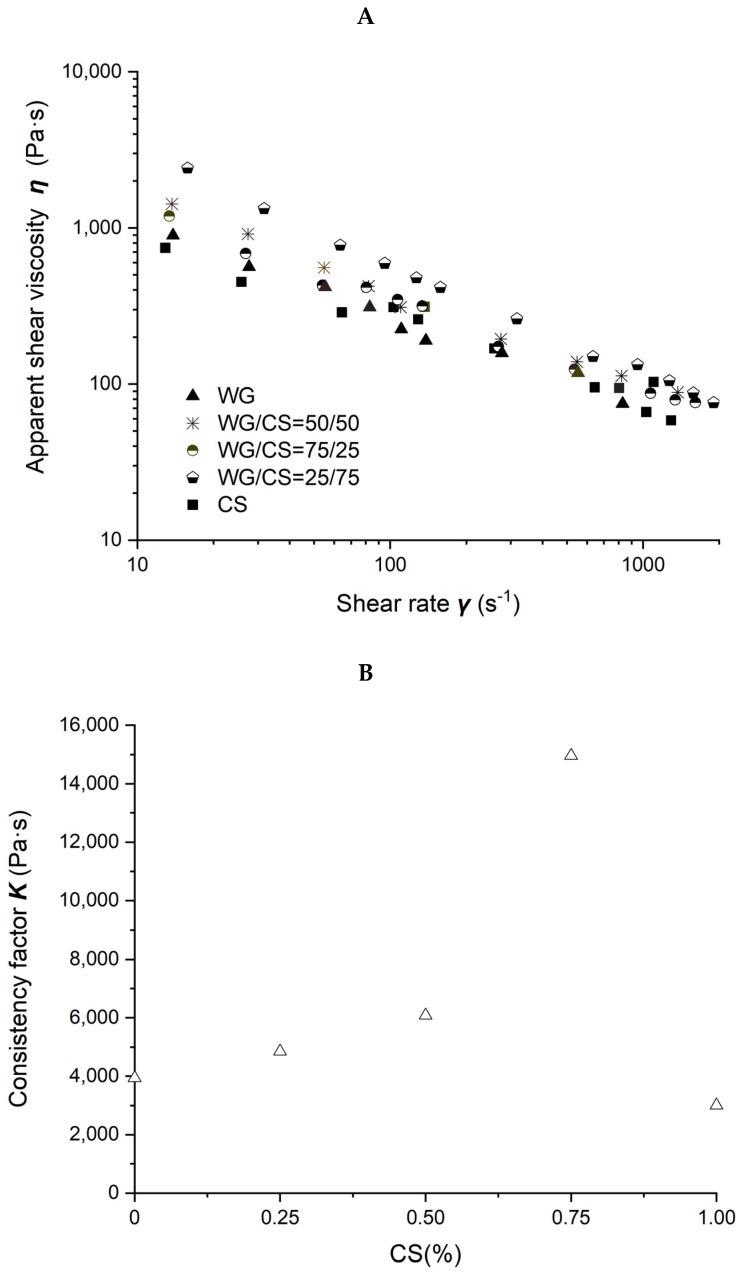
Rheological properties of wheat-gluten (WG)/corn-starch (CS) blends. Temperature was 80 °C and water content at 50% *w*/*w*. (**A**). Apparent shear viscosity; (**B**) consistency factor versus CS content in WG/CS blends; (**C**) viscosity exponent versus CS content in WG/CS blends.

**Figure 5 molecules-27-06693-f005:**
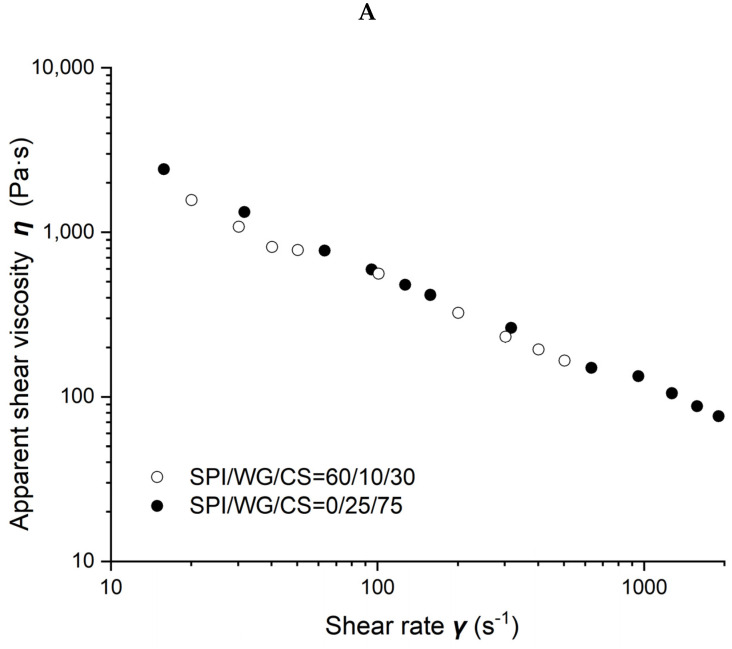
Rheological properties of soybean–protein–isolate (SPI)/wheat–gluten (WG)/corn–starch (CS) blends. Temperature was 80 °C and moisture was 50%. (**A**). Apparent shear viscosity of WG:CS = 1:3 in blends; (**B**) apparent shear viscosity of WG:CS = 1:1 in blends; (**C**) apparent shear viscosity of WG:CS = 3:1 in blends; (**D**) consistency factor versus SPI content in SPI/WG/CS blends; (**E**) viscosity exponent versus SPI content in SPI/WG/CS blends.

**Table 1 molecules-27-06693-t001:** Thermal transition properties of SPI, CS, and WG.

Materials	T_o_/°C	T_p_/°C	ΔH/J g^−1^
SPI	103.11 ± 0.92	108.25 ± 0.86	4.45 ± 0.37
CS	63.54 ± 0.61	68.56 ± 0.40	3.62 ± 0.61
WG	--	--	--

**Table 2 molecules-27-06693-t002:** The fitted equation of the viscosity exponent *n_no-inter_* of a non-interactive model for blends of soybean–protein–isolate (SPI)/wheat-gluten (WG)/corn-starch (CS).

Samples	X	Linear Fitting Equation	R^2^	Exponential Fitting Equation	R^2^	Power Function Fitting Equation	Logarithm Fitting Equation
SPI/CS	CS%	*n* = 0.1424x + 0.2666	0.625	*n* = 0.2744e^0.2852x^	0.652	\	\
SPI/WG	WG%	*n* = 0.1044x + 0.2748	0.631	*n* = 0.2793e^0.2974x^	0.652	\	\
WG/CS	CS%	*n* = 0.0464x + 0.4184	0.997	*n* = 0.4187e^0.105x^	0.999	\	\
SPI-WG/CS = 1/3	SPI%	*n* = −0.1625x + 0.437	0.862	*n* = 0.4344e^−0.440x^	0.867	\	\
SPI-WG/CS = 1/1	SPI%	*n* = −0.1536x + 0.4271	0.881	*n* = 0.4249e^−0.421x^	0.885	\	\
SPI-WG/CS = 3/1	SPI%	*n* = −0.1393x + 0.4173	0.866	*n* = 0.4154e^−0.387x^	0.870	\	\

**Table 3 molecules-27-06693-t003:** The fitted equation of the consistency factor *K_no-inter_* of a non-interactive model for blends of soybean–protein–isolate (SPI)/wheat–gluten (WG)/corn–starch (CS).

Samples	X	Linear Fitting Equation	R^2^	Exponential Fitting Equation	R^2^	Power Function Fitting Equation	Logarithm Fitting Equation
SPI/CS	CS%	K = −116720x + 119148	0.9999	K = 194371e^−3.371x^	0.813	\	\
SPI/WG	WG%	K = −115708x + 119287	1.0000	K = 184951e^−3.143x^	0.828	\	\
WG/CS	CS%	K = −934.88x + 3928.5	0.9993	K = 3945e^−0.271x^	1.000	\	\
SPI-WG/CS = 1/1	SPI%	K = 115934x + 3309.6	1.0000	K = 4419.1e^3.7195x^	0.931	\	\
SPI-WG/CS = 1/3	SPI%	K = 116226x + 3132.7	1.0000	K = 4133.8e^3.7382x^	0.923	\	\
SPI-WG/CS = 3/1	SPI%	K = 115736x + 3556.7	1.000	K = 4667.1e^3.5954x^	0.926	\	\

**Table 4 molecules-27-06693-t004:** Blends of soybean–protein–isolate (SPI)/wheat–gluten (WG)/corn–starch (CS) at different mass fractions.

Samples	SPI(%)	WG(%)	CS(%)
1	100	0	0
2	0	100	0
3	0	0	100
4	75	25	0
5	50	50	0
6	25	75	0
7	75	0	25
8	50	0	50
9	25	0	75
10	0	75	25
11	0	50	50
12	0	25	75
13	60	10	30
14	60	20	20
15	60	30	10
16	80	10	10

## Data Availability

The data presented in this study are available on request from the corresponding author.

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
