# Peer review of "The Consistency Factor and the Viscosity Exponent of Soybean-Protein-Isolate/Wheat-Gluten/Corn-Starch Blends by Using a Capillary Rheometry"

_molecules, 2022, doi:10.3390/molecules27196693_

Round 1

Reviewer 1 Report

It is recommended to use the volume fractions in equation 2, considering that the dynamic viscosity is a parameter that characterizes the flow of a fluid, rather of a concentrated solid-liquid suspension.

All figures are mixed in terms of notations A, B, C.

There is no motivation for choosing the added moisture content, respectively for the temperature of the rheo viscosimeter cylinder. In the case of temperature, the use of the 2 temperatures for the phase transition of 2 out of three components of the mixture, are not sufficient to state a theory and choose a working temperature.

The same in the case of mixture moisture. The explanations provided by the authors (lines 234-251) are not relevant for the choices made! In addition, in the conclusions (row 329), the moisture is presented as being below 50%, which denotes an inconsistency in complying with the imposed process parameters.

In the "Materials" section, it is not specified where or how the percentages of dry base and incorporated water were determined.

The symbols of the mixtures used are inconsistent: in one way they are presented in table 1 (numerical code), in another way in figures (symbols and percentages) and in another way in the text (symbols and ratios), which greatly complicates the comparison process of the data.

In addition, it is recommended that all the numerical results obtained from the statistical analysis be presented in tables (as in the case of tables 3 and 4), without any more numerical strings in the text - for a better clarity of the results.

Author Response

第1:建议使用等式2中的体积分数,因为动态粘度是表征流体流动的参数,而不是浓缩的固液悬浮液。

回应1感谢您的好建议,体积分数可能适合等式2。考虑到在流变测量过程中,在粘度计圆筒中很难获得SPI /WG/CS混合物的体积分数,并且根据参考[25],在等式2中使用了质量分数。

第2所有数字在符号A,B,C方面混合在一起。

回应2我们无法很好地理解评论。注释是否建议将“图 12345”修改为“图.“?如果是这样,则在作者的说明中没有对此要求。如果没有,审稿人能否更清楚地描述评论。多谢。

第3没有动机选择添加的水分含量,分别针对流变粘度计圆筒的温度。在温度的情况下,使用2个温度进行相变的2个混合物的三个组分,不足以陈述一个理论和选择一个工作温度。

回应3“根据桶中水分含量在40-60%的范围内,在植物基蛋白质的高水分挤出过程中,因此在所有测试中都选择了50%的水分含量。

“一般来说,在植物基蛋白质的高水分挤出过程中,桶温度高于100°C,这将导致水分快速消散,并在流变测量期间烘烤样品。因此,选择温度80°C,接近100°C并导致缓慢耗散。

第4在混合物水分的情况下也是如此。作者提供的解释(第234-251行)与所做的选择无关!此外,在结论(第329行)中,水分低于50%,这表明在遵守施加的工艺参数方面存在不一致。

回应4“根据桶中水分含量在40-60%的范围内,在高水分挤压过程中对植物基蛋白质进行挤压,因此在所有测试中都选择了50%的水分含量。

第329行被修改为“混合物在50%水分和80°C下表现出剪切变稀行为,粘度指数n范围为0.245至0.466”。

第5在“材料”部分,没有说明在何处或如何确定干碱和掺入水的百分比。

回应5将其修改为“将水添加到每个蛋白质 - 淀粉混合物中(作为干基总样品质量的一部分),并在食品炊具HR7633 / 10(荷兰飞利浦)中混合”。

第6所用混合物的符号不一致:它们以一种方式出现在表1(数字代码)中,以另一种方式出现在数字(符号和百分比)中,另一种方式出现在文本中(符号和比率),这大大增加了数据的比较过程。

响应6表1中数字代码的格式以及数字中的符号和百分比以及文本中的符号和比率已修改为一致。

第7此外,建议从统计分析中获得的所有数值结果都列在表格中(如表3和表4),案文中不再有数字字符串,以便使结果更加清楚。

答复7数字结果以数字形式呈现,这些数字比表格更直观。因此,建议不要进行任何修改。

Reviewer 2 Report

The manuscript ID molecules-1944398, entitled " The consistency factor and the viscosity exponent of soybean- protein-isolate/wheat-gluten/corn-starch blends by using a capillary rheometry" is well written and presents well designed experiment.

The Introduction section include an adequately argumentation of the research goal.

Material section provide key details on methods used.

The results and the discussions were detailed and were lead very well with board discussion with literature. I recommend to complete the discussion section with recent literature (within last 5 years).

Conclusions are properly drawn.

Provided literature is relevant to the research but it needs to be complete with literature published mostly within last 5 years.

Other remarks and comments:

Line 20: The coefficient of determination … instead of R2 ….

Line 55: … and works in a quadratic polynomial equation …. Please revised the formulation! 

Line 97: R2=-0.872 instead of R2=0.872

Line 109: R2=- 0.575 instead of R2= 0.575

Line 205: soybean instead of so-ybean

Line 224: … almost 1.000. instead of … almost 1.000,

Line 296: …. all the coefficients of determination (R2) of fitting models are …. instead of …. all the R2 of fitting …

Line 333: The coefficient of determination of the linear relationship …. instead of R2 of the linear relationship ….

Author Response

Point 1: Line 20: The coefficient of determination … instead of R2 ….

Response 1: Line 20 is revised into “ The coefficient of determination ”.

Point 2: Line 55: … and works in a quadratic polynomial equation …. Please revised the formulation! 

Response 2: Line 55 is revised into “...and is quadratically correlated with n value...”.

Point 3: Line 97: R2=-0.872 instead of R2=0.872

Response 3: The coefficient of determination R2 should not be negative. If the “-0.872” represent negative correlation, the meaning was described in the text as “negative linear correlation”.

Point 4: Line 109: R2=- 0.575 instead of R2= 0.575

Response 4: The coefficient of determination R2 should not be negative. If the “-0.575” represent negative correlation, the meaning was described in the text as “negative linear correlation”.

Point 5: Line 205: soybean instead of so-ybean

Response 5: so-ybean is revised into “ soybean ”.

Point 6: Line 224: … almost 1.000. instead of … almost 1.000,

Response 6: almost 1.000, is revised into “ … almost 1.000. ”.

Point 7: Line 296: …. all the coefficients of determination (R2) of fitting models are …. instead of …. all the R2 of fitting …

Response 7: Line 296 is revised into “ …. all the coefficients of determination (R2) of fitting models are …. ”.

Point 8: Line 333: The coefficient of determination of the linear relationship …. instead of R2 of the linear relationship ….

Response 8: Line 333 is revised into “ The coefficient of determination of the linear relationship …. ”.

Round 2

Reviewer 1 Report

NONE